# Knowledge, Use and Attitude of Information and Communication Technologies (ICTs) in Graduate Nursing Students: A Correlational Cross-Sectional Study

**DOI:** 10.3390/healthcare11141989

**Published:** 2023-07-10

**Authors:** Alberto Cruz-Barrientos, Ines Carmona-Barrientos, Jose Manuel De-la-Fuente-Rodriguez, Veronica Perez-Cabezas, Gloria Gonzalez-Medina, Ana Maria Sainz-Otero

**Affiliations:** 1‘Salus Infirmorum’ University Nursing Centre, University of Cadiz, 11001 Cadiz, Spain; 2Department of Nursing and Physiotherapy, University of Cadiz, 11009 Cadiz, Spain

**Keywords:** knowledge, use, attitude, ICT, nursing

## Abstract

Education in the XXI century is called to move forward in the right direction and to gain momentum to face diverse challenges and take opportunities offered by the knowledge that is inherent in society. Therefore, it can be postulated that there must be a close relationship between learning, the generation of knowledge, continuous innovation and the use of new technologies. This is a cross-sectional descriptive study conducted among a sample of 242 first year and second year Cadiz University nursing students, using the validated Learning and Study Strategies Inventory questionnaire to assess motivation and the Relationship between Learning Styles and Information and Communication Technologies questionnaire to assess knowledge, use and attitude. The predominant profile type of nursing degree students was women under 21 years old, who were single and exclusively dedicated to their studies. In addition, they showed positive motivation in relation to learning while facing adversity (between 76% and 76.6%). Regarding the level of knowledge, use and attitude towards ICTs, most of the considered items presented high values. For some questions, age was proven to be a sociodemographic variable that influenced both the motivation and attitude of the students. The level of knowledge, use and attitude that students have regarding ICTs are good when we refer to communication technologies, interpersonal relationships or basic programs. However, they are low when considering programs related to education or learning.

## 1. Introduction

Currently, society is immersed in the so-called postmodern era, which makes it necessary to redefine the quality of education together with the use of new methodological approaches, tools and teaching strategies that give prominence to the student as an active individual. The articulation of information and communication technologies (ICTs) significantly influences the learning process, the ways of relating to society and the management of knowledge itself. The United Nations (UN), through the United Nations Development Programme (UNDP), placed information and communication technologies for health (ICTs) at the service of human development, which were established over time in Latin American and European countries through their respective national government plans [1].

While higher education has benefited from technological tools, educational processes were supported to advance the levels of understanding and management of ICTs; the same ones that are used today to enable better levels of learning for students and professionals to generate the skills and abilities to gather information, process it and use it to engage in science and research and, therefore, strengthen the construction of responsible citizens who are capable of having active roles in public deliberation. Over two decades, university models have been generated according to their degree of technological development in organizational and educational aspects, teaching practice and type of leadership, differing from each other by the following factors:Leadership (with high integration of ICTs, regular teaching practice and driving leadership in ICTs).Cooperation (with high integration of ICTs, less presence in continuous training, skepticism in vocational matters and driving leadership).Self-sufficiency (with discreet integration of ICTs, teaching skepticism and lack of driving leadership).Skepticism (with no integration of ICTs, skeptical teachers and absence of driving leadership).Undertaking improvement actions to achieve a high integration of ICTs, therefore, improves training processes and teaching practices together with institutional support [2].

Within the framework of the situational diagnosis in the institutional sphere and of all careers, a nursing career is related to other academic units in terms of the following basic characteristics of ICTs: the application and knowledge of the institution’s contribution to ICTs and the current educational model being used in order to generate an analysis of strengths, weaknesses, threats and opportunities; in reference to the teaching field and the curricular field, the basis for the development of the research objective that starts from the perception of a deficit in the methodological use of technological resources or tools; and the weak integration of technological professional content in the nursing curriculum.

Social changes are produced by the impulse to obtain new knowledge and new technologies that societies develop. The technological capital of each of them determines their rate of change, and sociologists speak of comparing them according to their distinctive technology [3]. The technologies of the contemporary world are made up of information and communication technologies (ICTs). These, by facilitating the manipulation of information and providing the means for the acquisition, production, storage, communication, recording, presentation and transmission of data in previously unimaginable quantities and by facilitating communication between people, have transformed the social world [4]. Traditional literacy is insufficient in the network society of the information age [5]. Even the basic use of a computer, the telephone or any of the technologies of post-industrial society is not enough. Today, there is a requirement for information literacy that transcends the mere existence of information and emphasizes how to search for, find, use and manipulate it so that it becomes individual and social knowledge [6]. Health systems around the world have reacted to these requirements of seeking new types of organization and management in the ways of providing health services, optimizing existing resources and providing better quality care [7].

The trends show advances in the incorporation of ICTs in clinical–administrative systems, the unique identification of people, remote medical appointments, electronic medical records and electronic medical prescriptions [8], just to cite examples that show the search for better care, greater coverage and continuity of care, the improvement of communication processes and a greater adequacy of available health resources for existing demands [7]. Nursing professionals have decidedly opted for the use of information and communication technologies, thinking about the benefits that they can bring not only to the provision of health services to the population through instruments such as digital medical records and telecare, but also as a tool for improving communication processes and the management of knowledge and research [7].

An encouraging sign in this regard is the proliferation of nursing blogs, whose content is aimed at offering nursing and health news in general, with objectives such as promoting communication and collective action; being a space for the exchange of knowledge, experiences and resources; promoting and offering training for the care of the elderly and dependents; and contributing to the digital development of citizenship. There are many scientific articles that urge nursing professionals to research as a way and process of development and professional growth [9], because it is essential at all levels of care and for all health professionals. However, few perform research and, in many cases, they lack adequate support and motivation [10,11]. ICTs have been introduced to counteract such a situation, since they facilitate communication in real time with people anywhere in the world, thus eliminating the barriers that previously existed, allowing access to updated information sources and strengthening the research and development of nursing knowledge [12]. Mastery of them strengthens this and any profession and reaffirms its scientific capacity [6,13] on any subject and becomes the greatest source of nursing knowledge that has ever existed. With all this information, it is now much easier to improve a technique, answer a question, create or update a protocol, deal with a problem and improve the care that can be offered to patients [14]. Despite all this, the level of knowledge of ICTs is still low and their use in health sciences still faces distrust, fear, resistance and little hope [15].

Regarding research, studies on the use and perceptions of ICTs in the area of nursing have also been growing. The work that is often cited as pioneering is a study of attitudes by Brodt and Stronge carried out in 1985 [16]. Another along the same lines, which explores attitudes and perceptions towards the use of electronic medical records, is that by Moody et al. [17]. However, probably the most extensive and paradigmatic of its kind is that carried out by the Australian Nursing Federation and financed by the Australian government, in which ten thousand nurses in the country were contacted. This work, with a sample of 4400 nurses, provided reliable information about how nurses recognized the benefits of adopting ICTs in the workplace and their frustration due to limited access, the lack of better adaptation of the software and the lack of opportunities for better training in the area [18]. In particular, information skills are a set of skills, knowledge and attitudes to access information and make use of it. ICT-related skills are those that “allow the acquisition of relevant information effectively and efficiently, with the most appropriate means and channels in each case; (…) those related to the symbolic analysis of information, critical thinking and the ability to generate knowledge from information, its analysis and experience, and those related to self-development: the ability to learn to learn, the ability to innovate and learn from mistakes” [19].

The objective of this study was to determine the level of knowledge, attitudes, levels of use and motivations of undergraduate nursing students at the University of Cadiz in relation to ICTs, analyze the possible existence of some sociodemographic variables that may influence or be related to learning motivations and to the knowledge, attitudes and use of ICT and to propose improvements to the use of nursing informatics tools that help in the teaching–learning process. The variables used were sociodemographic, such as age, gender, etc.

## 2. Materials and Methods

### 2.1. Sample

The methodological approach is of a quantitative type, and this is a transversal descriptive study.

Following the criteria of efficiency, the sampling method used was simple sampling, not probabilistic, for convenience.

The target population selected considered first-year and second-year Cadiz University Nursing Degree students, consisting of students of that are part of the Cadiz, Jerez, Algeciras campus as well as the ‘Salus Infirmorum’ University Nursing Centre, which is both a part of Cadiz University and is within the Cadiz campus. Out of a total of 660 students belonging to the first and second year of the nursing degree, a final sample of 242 students was obtained. The criteria that defined the participation of the subjects were students who wanted to participate in the study and gave their consent, first and second year students of a Nursing Degree of the University of Cadiz and students who were in class at the time of the questionnaire.

### 2.2. Evaluation Questionnaire

As data collection for this study, the following instruments were used:A self-created recording sheet, where questions about sociodemographic variables and those related to the Nursing Degree were collected. We established the profile of the sample, i.e., the age, sex, marital status, province, city, profession, campus, course, subject and the system through which they were granted access to the university.The LASSI questionnaire [20] assessed the motivation of students towards learning. It was developed as part of the Cognitive Project of Strategy Learning at the University of Texas, Austin, and consists of 10 subscales with a total of 77 items. Its reliability and validity have been verified in the Spanish population [21]. Similarly, it shows good psychometric properties, obtaining a Cronbach’s Alpha coefficient of 0.72. The “Motivation” scale measures the degree to which students accept their responsibility to perform specific tasks related to academic success, as well as the desire and energy displayed when carrying out a specific study task.

A Likert-type scale was used to score the following: It is not my characteristic (0), It is not a very common feature in me (1), It is either partly or a little characteristic of me (2), It is often a characteristic of me (3) and It is something very characteristic in me (4).

3.The validated questionnaire Relationship between Learning Styles and Information and Communication Technologies [22] was also applied. This questionnaire was created at the Faculty of Educational Sciences of Albacete, to assess the knowledge, use and attitude that students have towards ICT and for how ICT is used by ICT students according to their learning style. After the final review, the Relationship between Learning Styles and Information and Communication Technologies questionnaire was sent to a group of experts for validation and a pilot test, showing a high reliability with a Cronbach’s Alpha of 0.852.

The questionnaire consists of 60 Likert-type items, from 1 (not at all) to 4 (a lot), and is divided into four subgroups: knowledge (items 1–14); use (items 15–28); attitude (items 29–44); and use of ICT according to the learning style (items 45–60) following the classification of Alonso, Gallego and Honey [23]. For our study, we used the subgroups of knowledge, use and attitude.

### 2.3. Data Collection

From November 2016 to February 2017, data collection was carried out among the Nursing Degree students in all the abovementioned areas of Cadiz University.

This project complies with all the ethical requirements established by the Declaration of Helsinki. All participants were properly informed about the purposes of the study and signed an informed consent to participate in the project.

A questionnaire in paper format was used to complete the Test Battery or through the computer platform “Survey Monkey”. There were no conditions on face-to-face or on-line data collection. It was left to the student’s choice.

### 2.4. Statistical Analysis

For data processing, we used the statistical package “SPSS 21” (“Statistical Package for the Social Sciences”) for Windows and the 2016 Office Suite.

For statistical analyses of coded data, a descriptive analysis of all the collected variables was first carried out. For this, frequencies and percentages were calculated for qualitative variables. The average, typical deviation and both the minimum and maximum were worked out for quantitative variables.

To analyze relationships between variables, the normality of quantitative variables was first confirmed, using the Kolmogorov–Smirnov test. Once this was obtained, it was shown that all variables followed a non-normal distribution and nonparametric tests were carried out.

To analyze differences between sociodemographic variables, mean comparisons were carried out using Mann–Whitney U tests for dichotomous variables and Kruskal–Wallis H in more than two categories. To analyze the relationship between age and scale, the Spearman correlation coefficients were calculated.

In all the analyses, a significance level *p*-value < 0.05 was considered.

## 3. Results

The predominant profile type of nursing students constituting our sample was women under 21 years old, single, with exclusive dedication to their studies, and their access to the university was mainly through an entrance exam.

In the results of the Motivation scale, we highlighted questions 2 and 4 “Even if the study materials are boring and without interest, I try to continue studying until the end” and “I study hard to get good qualifications, even if I do not like the subject” as being frequently or something very characteristic of students, with 76% and 76.8%, respectively.

Comparing sociodemographic variables with motivation, we found no significant differences except in the results of Spearman’s Rho with age, where we observed that there is an indirect correlation in questions 4, 5 and 6: “I study hard to get good qualifications, even if I don´t like the subject” (−0.161 (0.012)), “I often give some excuse for not doing the academic tasks” (−0.156 (0.015)) and “I set very high goals in my studies” (−0.150 (0.019)). Therefore, we can affirm that at a higher age, the learning motivation of the students is negative in relation to these questions (Table 1).

Analyzing the percentages of ICT knowledge, items 10, 14 and 16 were highlighted and relate to “I know personal interrelation programs, online search programs or online video portals”. Nursing students answered that they knew them pretty much or a lot with 91.7%, 88% and 90.9%, respectively. Unlike the discrete scores collected in questions 20 and 21 about the knowledge of educational tools, students answered “Author’s educational programs” and “Guided internet search activities” with nothing or something at 84.9% and 88.8%, respectively (Table 2).

In the analysis of results of use of ICTs, although no significant results were obtained, it was generally observed that students use basic and personal interrelation programs (items 23 and 24) quite or pretty much, at 91.7%. On the other hand, when we consider author educational programs (item 33) or guided internet search activities (item 34), they use them little or almost not at all (90%) (Table 3).

Regarding the attitude of students with respect to ICT, there are six significant items to which the students answered quite or much with percentages above 80%. These are “ICTs are an important part of my academic training” (item 36), “ICTs help me in my learning process” (item 37), “ICTs are important for their educational application” (item 39), “ICTs are a support to complete my academic and formative knowledge” (item 43), “ICTs are helpful to search for information” (item 48) and “ICTs are a useful tool for the preparation of homework” (item 49).

Despite this favorable attitude, slightly more than half of the students were not satisfied when we replace the traditional model of education with new technologies, as observed in item 46. “ICTs do not replace traditional educational resources”, where students answered nothing or something with an exact percentage of 56.9%. “ICTs hurt me more than they help me in my academic training” (item 38) was answered with nothing or something at 77.4% and “ICTs are difficult to understand and use” (item 42) was answered by 75.2% with nothing or something.

We found several questions with indirect correlations with age (items 38, 41, 43 and 51): “ICTs hurt me more than they help me in my academic training” −0.137 (0.035). Thus, the higher the age, the less favorable the students’ attitude towards whether ICTs are prejudicial in their academic education. “ICTs are a way to promote personal relationships among my classmates” −0.130 (0.045). At the same time, the results are inversely proportional and the higher the age considered, the less ICTs are seen as a way to promote relationships between students. In other words, “ICTs are a support to complete my academic and training knowledge” −0.161 (0.012). Hence, we can also say that the younger the age of students, the more favorable their attitude towards whether ICTs are a support for their academic and training knowledge. “ICTs help me in my free and leisure time”, so the older the age of the students, the less favorable their attitude to using ICTs in their free and leisure time (Table 4).

## 4. Discussion

When we interpret our work, the present study reveals that although students show a positive motivation towards learning (even in spite of adversities), the knowledge, use and attitude they have regarding new technologies is different when we talk about ICTs in the general domain and those related to the academic environment, with age being a crucial variable in these results.

### 4.1. Motivation

The results of this study show that although students face some adversity in the learning process, like the study materials are monotonous or the subject is not pleasant, the motivation shown by the students is positive enough to facilitate the learning process. Keeping in mind that motivation is one of the engines that drives learning and that nowadays students have a broader mastery of new technologies compared to adults, ICTs can be quite stimulating and motivating and can be transformed into ideal instruments to respond to the new type of students. This was reflected in [24,25,26,27,28,29], where students showed higher levels of motivation and using new technologies was proven to deliver a more effective way of learning.

Based on the results obtained, we can deduce that for these students, difficult problems are not an obstacle, but rather a challenge. They are sure that making an effort is the basis of success or failure. Therefore, we can say that students in general are quite motivated. These results are related to other investigations [30,31], where the authors concluded that students who participate in ICT strategies, such as trying to overcome challenges or get badges, are more motivated to complete learning tasks.

### 4.2. Knowledge

There are significant differences in the knowledge that students have of ICTs in general and those related to virtual or educational platforms.

On one hand, in questions related to social networks, nursing students said they knew enough or a lot about these technologies, but very little about the educational field. This coincides with some other studies carried out [32,33], where students at the Faculty of Education in Albacete reported little knowledge about web page editors, author educational programs and guided activities on the internet, but a high level of knowledge with respect to social network technologies used for communication, such as Messenger, Facebook or email.

The data we collected may have arisen due to the fact that students have a lot of knowledge and many skills about the internet and all kinds of electronic devices such as computers, mobile devices and PDAs, among others used socially and daily, and almost none on instruments for learning or educational tools. This coincides with previously published studies. Our work shows results very similar to those in [34,35].

The discrete scores collected in knowledge questions of educational tools are relevant. These data agree with a study [36] where they analyzed a group of second-year teaching students at the University of Girona on their knowledge and use of ICTs. It was observed that the knowledge they had of social networks, text processors or applications that generate slides was high, but knowledge about webquests or educational programs such as J Clic was limited.

Although students are easily able to explore the necessary technological tools, they claim not to know them. As a future objective to work on, it is proposed to encourage the carrying out of training courses by different, relevant departments within the university.

### 4.3. Use

When interpreting the use of the programs for personal reasons, the students reported a lot of use, but little in the way of educational use. In this sense, the research carried out [37] at the University of Murcia among 487 students confirms the hypothesis that the majority of students use these tools in a merely playful way, but with very little educational use. In another case study carried out by Gómez, Roses and Farias, university students recognized that the academic use of these tools is quite scarce. In the same line, we find other publications that even highlight the damages of the misuse of technologies [38,39,40].

Likewise, there are matching results from the study of Subires and Olmedo, Gillissen et al. and Webb et al. [41,42,43], whose students affirmed that they used social networks mostly to interact with peers, but they also valued that they could use them for an academic–professional purpose as long as it facilitated their work in the class.

### 4.4. Attitude

Regarding the attitude that students have towards ICTs, this study shows a positive general attitude. This was also the case in several studies present in the scientific literature, such as that by Gillissen et al. [42], where the majority of medical students expressed positive attitudes towards digital applications in medicine. The review of fifty selected studies is very interesting [43]. This review clarifies more advantages than before for ICT in nursing education and it is supported by the advantage of flexibility. Additionally, in work by Moreno and Delgado [44], students of Psychopedagogy from the University of Extremadura, whose results showed that a significant number of students had a preference for and a positive attitude towards a type of didactic methodology based on ICTs in comparison with a traditional or master class.

Opposite to the results obtained are the studies conducted by Shaw and Marlow and Garrote et al. [40,45], which showed that students of Nutrition and Sports Science had a negative attitude towards new technologies, since they did not feel comfortable with computers and they were not happy with the lack of direct contact and the dangers caused by the abuse of ICT.

It should be noted that in the statement “ICTs do not replace traditional educational resources”, the majority answered nothing or something. So, we could say that despite their favorable attitude, half of the students do not agree with replacing the traditional model of education with new technologies. This result is very similar to those of Bloomfield et al. [46], who compared traditional learning with learning using new technologies among nursing students. They came to the conclusion that students had the same attitude regardless of the learning model. In the same line, in an interview with medical students in Germany, it was concluded that despite having a favorable attitude towards ICTs, the majority did not agree with substituting new technologies for the traditional educational model [42].

Likewise, we observed that the older the students, the more negative the attitude students have towards ICTs. This observation is contrary to the study carried out by Padilla et al. [38] at the University of Seville among students over 25, 40 and 45 years old who accessed the university, which returned a generally positive attitude towards ICTs based on their potential benefits in learning processes. ICTs are part of the daily and working life of healthcare professionals. In relation to education within the nursing degree, ICTs make it possible to carry out practical immersion training from the classroom or from any other place with an internet connection, which has become evident in the circumstances that have occurred in recent years, such as those as a result of the pandemic caused by COVID-19 [47,48,49,50].

It is important to note that since the COVID-19 pandemic, there has been a clear trend towards an increase in the use of ICTs and an improvement in the attitude towards them [51].

## 5. Conclusions

Nursing students from Cadiz University have a good level of motivation towards learning, even in cases that are not favorable for learning, such as tedious subjects or those with a very difficult content.

The other variables analyzed, such as the level of knowledge of, use of and attitude towards ICTs, show high values when we refer to communication technologies, interpersonal relationships or basic programs. However, the values are quite low when we consider programs related to education or learning.

Regarding the sociodemographic characteristics and the relationship between learning motivation, knowledge, use and attitude and ICT, there are no significant differences regarding the variables of gender, marital status or university campus. However, there is a difference in the age variable.

Due to the results obtained in terms of the lack of knowledge and use of educational tools by students, we thought it would be necessary to promote proposals for improvement to aid the most relevant teaching–learning process, such as bringing technology into a greater number of teaching activities as an educational tool that benefits the learning process of undergraduate nursing students.

ICTs should be integrated into the classroom and both become a cognitive instrument and encourage students to learn and develop their skills.

## Figures and Tables

**Table 1 healthcare-11-01989-t001:** Descriptive analysis of the motivation towards ICT (LASSI Scale) and correlation with age.

Variables	It Is Not My Characteristic	It Is Not a Very Common Feature in Me	It Is Either Partly or a Little Characteristic of Me	It Is Often a Characteristic of Mine	It Is Something Very Characteristic in Me	Rank 1–5	Mean (SD)	Correlation Coefficient (*p*-Value)
I’m up to date on my academic assignments	9 (3.7%)	23 (9.5%)	40 (16.5%)	108 (44.6%)	62 (25.6%)	1–5	3.79 (1.047)	−0.088 (0.171)
Even when the study materials are boring and without interest, I try to continue studying until the end	2 (0.8%)	11 (4.5%)	45 (18.6%)	107 (44.2%)	77 (31.8%)	1–5	4.02 (0.874)	−0.118 (0.067)
I come to class without preparing for it	21 (8.7%)	41 (16.9%)	77 (31.8%)	59 (24.4%)	44 (18.2%)	1–5	3.26 (1.193)	−0.002 (0.980)
I study hard to get good qualifications, even if I do not like the subject		6 (2.5%)	50 (20.7%)	100 (41.3%)	86 (35.5%)	2–5	4.10 (0.809)	−0.161 (0.012)
I often give some excuse for not doing the academic tasks	81 (33.5%)	76 (31.4%)	51 (21.1%)	24 (9.9%)	10 (4.1%)	1–5	2.20 (1.131)	−0.156 (0.015)
I set very high goals in my studies	14 (5.8%)	30 (12.4%)	45 (18.6%)	81 (33.5%)	72 (29.8%)	1–5	3.69 (1.187)	−0.150 (0.019)
When the study is difficult, I stop doing it or study only the easy parts	100 (41.3%)	76 (31.4%)	43 (17.8%)	18 (7.4%)	5 (2.1%)	1–5	1.98 (1.038)	−0.081 (0.208)
I read the textbooks indicated by the Professor	65 (26.9%)	80 (33.1%)	66 (27.3%)	22 (9.1%)	9 (3.7%)	1–5	2.30 (1.075)	0.007 (0.909)

**Table 2 healthcare-11-01989-t002:** Frequencies and percentages, means and standard deviations of the REATIC Scale on Knowledge of ICT.

Items	Nothing	Something	Quite	A Lot	Rank 1–4	Mean (SD)
Basic programs such as word processors (Word), spreadsheet (Excel), slide presentation (PowerPoint)	1 (0.4%)	30 (12.4%)	109 (45.0%)	102 (42.1%)	1–4	3.29 (0.693)
Personal interrelation programs (messenger, email, Tuenti, Facebook, Hi5)	4 (1.7%)	16 (6.7%)	84 (35.0%)	136 (56.7%)	1–4	3.47 (0.696)
What is a blog, a chat, a forum	12 (5.0%)	58 (24.1%)	93 (38.6%)	78 (32.4%)	1–4	2.98 (0.875)
Educational portals (RedCampus, Moodle, Webct)	45 (18.6%)	81 (33.5%)	78 (32.2%)	38 (15.7%)	1–4	2.45 (0.968)
Image editing programs (Paint, PhotoShop), video (Windows Media Maker, Pinnacle, Adobe Premier), audio (Windows Media, Winamp)	32 (13.3%)	84 (34.9%)	75 (31.1%)	50 (20.7%)	1–4	2.59 (0.962)
Online search programs (Google, Yahoo, Altavista)	6 (2.5%)	23 (9.5%)	96 (39.7%)	117 (48.3%)	1–4	3.34 (0.752)
Online translators (elmundo.es)	9 (3.7%)	43 (17.8%)	94 (39.0%)	95 (39.4%)	1–4	3.14 (0.840)
Online video portals (YouTube)	4 (1.7%)	18 (7.5%)	73 (30.3%)	146 (60.6%)	1–4	3.50 (0.708)
Libraries and virtual encyclopedias (Wikipedia, Encarta, Royal Academy of Language, Miguel de Cervantes)	6 (2.5%)	57 (23.7%)	108 (44.8%)	70 (29.0%)	1–4	3.00 (0.793)
Editors to make web pages (Frontpage, Dreamweaver)	134 (55.6%)	64 (26.6%)	34 (14.1%)	9 (3.7%)	1–4	1.66 (0.857)
Some web browsers (Explorer, Mozilla, Firefox, Netscape)	7 (2.9%)	49 (20.5%)	106 (44.4%)	77 (32.2%)	1–4	3.06 (0.802)
Author’s educational programs (Clic, JClic, Hot Potatoes, Neobook)	120 (50.2%)	83 (34.7%)	30 (12.6%)	6 (2.5%)	1–4	1.67 (0.790)
Guided internet search activities (Webquest, Miniwebquest, Hunt treasure)	131 (54.4%)	83 (34.4%)	22 (9.1%)	5 (2.1%)	1–4	1.59 (0.743)
Multimedia devices (PC, projector, Pda, scanner, Webcam)	39 (16.2%)	68 (28.2%)	84 (34.9%)	50 (20.7%)	1–4	2.60 (0.991)

**Table 3 healthcare-11-01989-t003:** Frequencies, percentages, means and standard deviations of the REATIC Scale on the Use of ICT.

Items	Nothing	Something	Quite	A Lot	Rank	Mean (SD)
Basic programs such as word processors (Word), spreadsheets (Excel), slide presentations (PowerPoint)		20 (8.3%)	92 (38.0%)	130 (53.7%)	2–4	3.45 (0.644)
Personal interrelation programs (messenger, email, Tuenti, Facebook, Hi5)	3 (1.2%)	17 (7.1%)	73 (30.3%)	148 (61.4%)	1–4	3.52 (0.684)
Educational portals (RedCampus, Moodle, Webct)	46 (19.0%)	82 (33.9%)	70 (28.9%)	44 (18.2%)	1–4	2.46 (0.998)
Image editing programs (Paint, Photoshop), video (Windows Media Maker, Pinnacle, Adobe Premier), audio (Windows Media, Winamp)	45 (18.8%)	90 (37.7%)	67 (28.0%)	37 (15.5%)	1–4	2.40 (0.965)
Online search programs (Google, Yahoo, Altavista)	2 (0.8%)	31 (12.8%)	86 (35.5%)	123 (50.8%)	1–4	3.36 (0.734)
Online translators (elmundo.es)	21 (8.8%)	66 (27.5%)	87 (36.3%)	66 (27.5%)	1–4	2.83 (0.934)
Online video portals (YouTube)	2 (0.8%)	31 (12.9%)	60 (25.0%)	147 (61.3%)	1–4	3.47 (0.748)
Libraries and virtual encyclopedias (Wikipedia, Encarta, Royal Academy of Language, Miguel de Cervantes)	10 (4.1%)	65 (26.9%)	109 (45.0%)	58 (24.0%)	1–4	2.89 (0.815)
Editors to make web pages (Frontpage, Dreamweaver)	147 (61.0%)	62 (25.7%)	23 (9.5%)	9 (3.7%)	1–4	1.56 (0.815)
Some web browsers (Explorer, Mozilla, Fire Fox, Netscape)	22 (9.1%)	63 (26.0%)	80 (33.1%)	77 (31.8%)	1–4	2.88 (0.965)
Author’s educational programs (Clic, JClic, Hot Potatoes, Neobook)	142 (58.7%)	71 (29.3%)	21 (8.7%)	8 (3.3%)	1–4	1.57 (0.787)
Guided internet search activities (Webquest, Miniwebquest, Hunt treasure)	141 (58.5%)	75 (31.1%)	16 (6.6%)	9 (3.7%)	1–4	1.56 (0.779)
Multimedia devices (PC, projector, Pda, scanner, Webcam)	43 (17.8%)	70 (28.9%)	78 (32.2%)	51 (21.1%)	1–4	2.57 (1.013)

**Table 4 healthcare-11-01989-t004:** Descriptive analysis of the attitude towards ICT (REATIC Scale) and correlation with age.

Items	Nothing	Something	Quite	A Lot	Rank	Mean (SD)	Correlation Coefficient (*p*-Value)
ICTs are an important element in my academic training		35 (14.5%)	104 (43.2%)	102 (42.3%)	2–4	3.28 (0.702)	−0.040 (0.475)
ICTs help me in my learning process	2 (0.8%)	41 (17.1%)	97 (40.4%)	100 (41.7%)	1–4	3.23 (0.755)	−0.079 (0.223)
ICTs hurt me more than help me in my academic training	93 (38.9%)	92 (38.5%)	38 (15.9%)	16 (6.7%)	1–4	1.90 (0.900)	−0.137 (0.035)
ICTs are important for their educational application	1 (0.4%)	44 (18.2%)	125 (51.7%)	72 (29.8%)	1–4	3.11 (0.697)	−0.045 (0.487)
ICTs helps me improve my academic results	8 (3.4%)	63 (26.8%)	96 (40.9%)	68 (28.9%)	1–4	2.95 (0.833)	−0.090 (0.169)
ICTs are a means to encourage personal relationships among my classmates	18 (7.5%)	64 (26.6%)	85 (35.3%)	74 (30.7%)	1–4	2.89 (0.929)	−0.130 (0.045)
ICTs are difficult to understand and use	53 (21.9%)	129 (53.3%)	46 (19.0%)	14 (5.8%)	1–4	2.09 (0.797)	0.010 (0.871)
ICTs are a support to complete my academic and formative knowledge	2 (0.8%)	36 (14.9%)	129 (53.3%)	75 (31.0%)	1–4	3.14 (0.688)	−0.161 (0.012)
ICTs do not offer me enough security in my privacy	51 (21.3%)	128 (53.3%)	46 (19.2%)	15 (6.3%)	1–4	2.10 (0.804)	−0.118 (0.067)
ICTs make me lose a lot of time	36 (14.9%)	133 (55.2%)	50 (20.7%)	22 (9.1%)	1–4	2.24 (0.817)	0.003 (0.969)
ICTs are not a substitute for traditional educational resources	28 (10.9%)	110 (46.0%)	76 (31.8%)	27 (11.3%)	1–4	2.44 (0.832)	−0.099 (0.128)
ICTs are essential in today’s society	4 (1.7%)	47 (19.4%)	99 (40.9%)	92 (38.0%)	1–4	3.15 (0.787)	−0.087 (0.176)
ICTs are a help to seek information		28 (11.7%)	99 (41.3%)	113 (47.1%)	2–4	3.35 (0.681)	−0.026 (0.685)
ICTs are a useful tool for the preparation of homework	2 (0.8%)	29 (12.1%)	89 (37.2%)	119 (49.8%)	1–4	3.36 (0.725)	−0.051 (0.436)
ICTs are not fully reliable in the information they provide	9 (3.7%)	135 (56.0%)	73 (30.3%)	24 (10.0%)	1–4	2.46 (0.724)	−0.101 (0.116)
ICTs help me to occupy my leisure and free time	19 (7.9%)	65 (26.9%)	83 (34.3%)	75 (31.0%)	1–4	2.88 (0.940)	−0.140 (0.030)

## Data Availability

Data presented in this study are not publicly available due to privacy restrictions.

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
