# Peer review of "Knowledge, Use and Attitude of Information and Communication Technologies (ICTs) in Graduate Nursing Students: A Correlational Cross-Sectional Study"

_healthcare, 2023, doi:10.3390/healthcare11141989_

Round 1
Reviewer 1 Report
First of all congratulations for the work done.
I believe that it may be a relevant work for certain researchers in the area of teaching and the use of ICT.
The introduction is extensive and theories are commented (Brunner's theoretical model and "a model for transformation) that are not subsequently discussed in magnitude in the discussion.
The methodology appears well exposed, perhaps it would be convenient to comment on the reference population in greater detail. As it is a convenience sample, this may be a limitation that should be discussed, perhaps explaining the sample well would be enough.
This study was carried out in 2016/17, more than 5 years ago. It is already said that the COVID-19 pandemic has increased the use of ICT. This is an important limitation in this field of study, where 5 years is a very long period, but also if something as special as COVID-19 has occurred, the results may not be current.
The same occurs with the cited bibliography.
Author Response
Dear Reviewer,
We sincerely appreciate your valuable comments ans suggestions on our article titled [Knowledge, use and attitude of Information and Communication Technologies (ICT) in nursing graduate stu-dents: a correlational cross-sectional study]. We highly value the time and effort you have dedicated to reviewing our work, and we are pleased to have opportunity to address each of the points raised
First of all congratulations for the work done.
I believe that it may be a relevant work for certain researchers in the area of teaching and the use of ICT.
Thanks for the comments
-
The introduction is extensive and theories are commented (Brunner's theoretical model and "a model for transformation) that are not subsequently discussed in magnitude in the discussion.
Dear Reviewer, thank you very much in advance. We appreciate your comments on our article. We have proceeded to corrected the paragraph In order to your observation. (lines 59-66) -
The methodology appears well exposed, perhaps it would be convenient to comment on the reference population in greater detail. As it is a convenience sample, this may be a limitation that should be discussed, perhaps explaining the sample well would be enough.
Thank you very much for your suggestion. We have proceeded to to clarify the point you raise. (lines 144-146)
-
This study was carried out in 2016/17, more than 5 years ago. It is already said that the COVID-19 pandemic has increased the use of ICT. This is an important limitation in this field of study, where 5 years is a very long period, but also if something as special as COVID-19 has occurred, the results may not be current.
Dear reviewer,
we sincerely appreciate your comments and suggestions on my article. we would like to address your concerns regarding the limitations of the publication year and the lack of consideration for the impact of COVID-19 on the use of ICTs.
First and foremost, we acknowledge that the publication year of my article is over five years. However, we believe that the information and analysis presented in my study remain relevant and useful for understanding the addressed topic. While technological advancements may have occurred since then, it is important to note that my main objective was to provide a broader theoretical or conceptual approach rather than focusing on the latest technological developments. Nonetheless, I can include a mention of the publication date and acknowledge the possible updates needed in future research.
Regarding your observation about the lack of consideration for the impact of COVID-19 on the use of ICTs, I find it to be a relevant point. Since my research was conducted prior to the pandemic, we did not have the opportunity to directly examine this impact in my study. However, we recognize that COVID-19 has driven significant changes in the use of ICTs in various domains, and I agree that it would be valuable to consider and discuss these implications in future research.
Taking your comments into account, we have committed to updating some of the bibliographic references in our article. (lines 363-367). This will enable readers to have a more comprehensive and up-to-date view of the topic.
Once again, I sincerely appreciate your insight and contribution to my work. Your comments are invaluable for improving the quality and relevance of our research.

Reviewer 2 Report
As an identified quantitative study, hypotheses, research questions, and dependent and independent variables, were not evident.
LINE 2, 14, 15: abbreviation: spell out the first time (ICMJE)
Abbreviations and Symbols: Use only standard abbreviations; the use of nonstandard abbreviations can be confusing to readers. Avoid abbreviations in the title of the manuscript. The spelled-out abbreviation followed by the abbreviation in parentheses should be used on the first mention unless the abbreviation is a standard unit of measurement. https://www.icmje.org/recommendations/browse/manuscript-preparation/preparing-for-submission.html#b
Line 22: replace bad
Line 29, 30, 31: redundant wording
Line 66-67: Anthropomorphism
Line 165 & 178: Consistency
Line 172-173: check abbreviation throughout the manuscript
Line 184: Under Process, was there any institutional approval?
Line 388-389: Confusing
Line 401: confidentiality or anonymity, but not both. (Anonymity means the identity is hidden from everyone, and Confidentiality means hiding the identity from the public. Confidentiality involves the agreement, while Anonymity doesn’t).
Minor grammar/sentence structure noted.
Author Response
Dear Reviewer,
We sincerely appreciate your valuable comments ans suggestions on our article titled [Knowledge, use and attitude of Information and Communication Technologies (ICT) in nursing graduate stu-dents: a correlational cross-sectional study]. We highly value the time and effort you have dedicated to reviewing our work, and we are pleased to have opportunity to address each of the points raised
-
As an identified quantitative study, hypotheses, research questions, and dependent and independent variables, were not evident.
Dear reviewer, we appreciate your comments on our article. In response to your observation regarding the hypotheses, since it is considered an observational descriptive study, formal hypotheses cannot be put forward, in the strict sense given to this concept in Experimental Methodology. Nevertheless, we do allow ourselves to outline the "Conceptual Hypothesis" that ICT in general, improve the teaching-learning process of the students of the Degree in Nursing.In relation to the research questions We have proceeded to add this sugestion (lines 128-133). Finally, in relation to the suggestion of the variables we have added it in the paragraph. (lines 133-134)
-
LINE 2, 14, 15: abbreviation: spell out the first time (ICMJE)
Abbreviations and Symbols: Use only standard abbreviations; the use of nonstandard abbreviations can be confusing to readers. Avoid abbreviations in the title of the manuscript. The spelled-out abbreviation followed by the abbreviation in parentheses should be used on the first mention unless the abbreviation is a standard unit of measurement. https://www.icmje.org/recommendations/browse/manuscript- preparation/preparing-for-submission.html#b
Thank you very much for your suggestion. We have proceeded to corrected, more details about abbrevation (lines 2-3) and (lines 15-17)
-
Line 22: Replace bad
Dear reviewer, We appreciate your comments on our article. In response to your observation of replace (line 24)
-
Line 29, 30, 31: redundant wording
Dear Reviewer, we have proceeded to rephrase the highlighted paragraphs in a clearer manner. Thank you very much for your input. (lines 29-32)
-
Line 66-67: Anthropomorphism
Thank you very much for your suggestion. We have proceeded to remove the phrase
1
-
Line 165 & 178: Consistency
Dear Reviewer, thank you very much in advance. We appreciate your comments on our article. We have proceeded to correct. (lines 160, 174)
-
Line 172-173: check abbreviation throughout the manuscript
Dear reviewer, We appreciate your comments on our article. The abbrevation have been modified. (lines 167-168)
-
Line 184: Under Process, was there any institutional approval?
Thank you for your comment and the opportunity to clarify the point you raise. we have proceeded to replace the Word (line 180)
-
Line 388-389: Confusing
Dear reviewer, We appreciate your comment in deed. We have proceeded to add this comment to the autor contributions (lines 389-399)
-
Line 401: confidentiality or anonymity, but not both. (Anonymity means the identity is hidden from everyone, and Confidentiality means hiding the identity from the public. Confidentiality involves the agreement, while Anonymity doesn’t).
Thank you very much for your suggestion. We have proceeded to to clarify the point you raise. (line 408-409)
11. Minor grammar/sentence structure noted.
Dear reviewer, thank you very much in advance for your comments. Unfortunately, the article has been translated by a translator, but we will take it into account for next time.

Reviewer 3 Report
the work is interesting and just to improve your understanding I suggest the following
The methodology must include the value of the population and specify the size of the sample based on the percentage reached, this is important to consider that it is sufficiently representative and the data can be generalized, as well as it is important to describe the criteria that defined the participation of the subjects.
I suggest establishing what the conditions were for those who completed the tests online, and if there were no implications or differences in the data collection.
In the scope of knowledge, the type of study must be corrected, it is not a descriptive study, it is a correlational study.
The precise statistic for this type of study is the multivariate correlation, check.
The scatterplots of the correlation should be added.
Author Response
Dear Reviewer,
We sincerely appreciate your valuable comments ans suggestions on our article titled [Knowledge, use and attitude of Information and Communication Technologies (ICT) in nursing graduate stu-dents: a correlational cross-sectional study]. We highly value the time and effort you have dedicated to reviewing our work, and we are pleased to have opportunity to address each of the points raised
The work is interesting and just to improve your understanding I suggest the following
Thank you very much for the comments.
- The methodology must include the value of the population and specify the size of the sample based on the percentage reached, this is important to consider that it is sufficiently representative and the data can be generalized, as well as it is important to describe the criteria that defined the participation of the subjects.
Dear Reviewer, Thank you in advance for your comment. We have proceeded to add this comment to the sample (lines 144-146)
- I suggest establishing what the conditions were for those who completed the tests online, and if there were no implications or differences in the data collection.
Thank you very much for your suggestion. We have proceeded to add more details about tests (lines 187-188)
- In the scope of knowledge, the type of study must be corrected, it is not a descriptive study, it is a correlational study.
Dear Reviewer, we have proceeded to rephrase the title (lines 3-4)
- The precise statistic for this type of study is the multivariate correlation, check.
Dear reviewer, We appreciate your comments on our article. In response to your observation regarding the precise statistic for this type of study, we would like to clarify that our main objective in this article was to determine the correlation with age, which is why we specifically conducted a bivariate analysis for that purpose.
In our study, we focused on examining the relationship between the variables in a bivariate approach, specifically in relation to age. Our aim was to explore whether there was a significant correlation between these variables. However, it is important to note that in the majority of our results, no significant correlations were found.
While we understand that a multivariate analysis could provide a more comprehensive perspective by considering other relevant variables, our intention was to specifically investigate and present the relationship between age and the variables analyzed.
We will take your suggestion into account, and in future research, we will consider the possibility of including a multivariate analysis to further explore the potential influences of other relevant variables on our findings.
We once again appreciate your comments and suggestions as they help us improve the quality and clarity of our work.
Our objective was only to know the correlation with age and that was the reason for the bivariate analysis. In addition, the correlations were not significant in most of the results.
- The scatterplots of the correlation should be added.
Thank you for your comments and the opportunity to clarify the point you raise. I think it is a good suggestion, but seeing that most of the correlations were not significant and those that were significant were low due to space limitations, we decided not to include them. If you think it is strictly necessary I will have no objection to put it. We appreciate your understanding and consideration of our decisión.
Thank you once again for your valuable feedback
